# COVID-19—Evidence of the Impact of Literacy and Salutogenic Behaviours in Positive Mental Health: A Cross-Sectional Study

**DOI:** 10.3390/bs13100845

**Published:** 2023-10-16

**Authors:** Cláudia Almeida, André Novo, Maria Lluch Canut, Carme Ferré-Grau, Carlos Sequeira

**Affiliations:** 1ULSNE—Unidade Local de Saúde Nordeste, 5301-852 Bragança, Portugal; 2Faculty of Nursing, Universitat Rovira y Virgili, 43003 Tarragona, Spain; carme.ferre@urv.cat; 3Instituto Politécnico de Bragança, Escola Superior de Saúde, 5300-121 Bragança, Portugal; andre@ipb.pt; 4Department of Public Health, Mental Health and Maternal and Child Health Nursing, Nursing School, University of Barcelona, 08007 Barcelona, Spain; tlluch@ub.edu; 5Escola Superior de Enfermagem do Porto, CINTESIS—Centro de Investigação em Tecnologias e Serviços de Saúde, 4200-072 Porto, Portugal; carlossequeira@esenf.pt

**Keywords:** mental health, literacy, COVID-19

## Abstract

Positive mental health is defined as the ability to perceive and interpret the context of a situation and to adapt to it whenever necessary. Considering the pandemic situation, identifying the factors that may have the greatest impact on quality of life and consequently, on positive mental health is paramount. The objective of this study was to assess the impact of health literacy on the adoption of behaviours that promote positive mental health during COVID-19. A descriptive, cross-sectional and correlational study was conducted on a sample of 770 patients using a questionnaire for sociodemographic characterization, the Positive Mental Health Questionnaire and the Mental Health Knowledge Questionnaire. Concerning health-promoting behaviours, those who sleep enough hours, exercise regularly, eat healthy and are more aware of mental health promotion activities, or have greater mental health literacy, have higher positive mental health scores. Thus, having more knowledge of mental health and adopting health-promoting behaviours improve positive mental health.

## 1. Introduction

In late December 2019, the first cases of atypical pneumonia of unknown origin were reported in China. In January 2020, Chinese authorities identified the infectious agent, naming it SARS-CoV-2, and the World Health Organization (WHO) declared the COVID-19 outbreak an international public health emergency [1]. On 18 March 2020, a state of emergency was declared in Portugal and consequently, several measures were implemented to protect public health.

Considering all the scientific evidence and the consequences that social isolation/quarantine periods may have on anxiety and stress, the WHO (2020) issued a statement aimed at minimizing these effects. The statement highlights the importance of exercising regularly, eating healthy and having sleep routines. Besides these factors, literacy was also pointed out as a key role player concerning the resources needed to minimize the impact of any circumstance, whether in a pandemic or a normal time, on an individual’s MH [2].

Several studies conducted over the past decade on different populations (students, institutionalized elderly people, psychiatric patients, young adults) indicate that factors such as healthy behaviours or health-promoting behaviours such as a good diet, frequent physical exercise, and good sleep hygiene have a positive impact on MH [3,4,5,6,7]. Similarly, other studies show that having a good level of positive mental health (PMH) is a protective and preventive factor against several mental disorders, including major depression, panic attacks and anxiety disorders [8]. Therefore, the need to foster PMH as a protective element of MH, both in the general population and among patients [8,9,10,11,12,13], has established itself as a predictor and/or mediator of adaptive reactions for managing situations generated by the pandemic and potential problems of MH such as suicide or addiction [13,14].

From a positive perspective of MH, the predominant model is the Dual-Continuous Model initially formulated by Keyes (2002) [15]. This model introduces an operationalization of PMH which is considered independently from mental illness, although they are interrelated. Lluch’s Multifactorial Model of Positive Mental Health [16] goes against this line of thought when formulating PMH as a construct that is defined by six interrelated factors: Factor 1: personal satisfaction (items 4, 6, 7, 12, 14, 31, 8 and 39); Factor 2: pro-social attitude (1, 3, 23, 25 and 37); Factor 3: self-control (2, 5, 21, 22 and 26); Factor 4: autonomy (10, 13, 19, 33 and 34); Factor 5: problem solving/self-maintenance (15, 16, 17, 27, 28, 29, 32, 35 and 36) and Factor 6: interpersonal relational skills (8, 9, 11, 18, 20, 24 and 30). 

Lluch’s Multifactorial Model of PMH (1999) is assessed using a standardized instrument which will be described in the methodology. This model allows for a salutogenic assessment of mental health and enables health professionals to be health-promoting agents by implementing interventions capable of promoting satisfaction, a pro-social attitude, self-control, autonomy, problem-solving skills and interpersonal relationships.

As far as mental disorders are concerned, they carry a great deal of weight when it comes to health in Europe, and their impact on quality of life is greater than chronic diseases, such as diabetes, and cardiovascular and respiratory diseases. Therefore, it is paramount not only to take care of MH problems and try to reduce their incidence, but also to provide the necessary resources to improve the MH of individuals who do not have MH problems [17]. We believe that developing specific intervention programmes to address MH problems requires a broad knowledge of how people react when faced with different situations and the factors that may be interrelated with such reactions.

In this regard, this study was proposed to identify which salutogenic behaviours can positively influence mental health and if mental health knowledge influences the positive mental health of individuals tested.

## 2. Materials and Methods

### 2.1. Design and Participants

This study was a descriptive, correlational and cross-sectional study. The opportunity sample was composed of 770 patients suspected of having a COVID-19 infection during the pandemic, who were screened at the testing units of a healthcare unit located in Northern Portugal and agreed to participate voluntarily. The data were collected during the lockdown phase of the pandemic and before vaccination.

### 2.2. Procedure

Inclusion criteria were individuals aged 18 or older who voluntarily signed an informed consent form and who had been tested for a suspected COVID-19 infection. Conversely, people who could not communicate in Portuguese or who did not have access to a computer with an internet connection to complete the assessment tools were not included in the study.

First, a form for free and informed consent to participate in the study was sent to each participant. After receiving the signed informed consent, the sociodemographic instrument, the Positive Mental Health Questionnaire (PMHQ) [18] and the Mental Health Knowledge Questionnaire (MHKQ) [19] were sent to the participants. Every instrument was delivered by email and completed on an online platform. The study took place from October 2020 to January 2021.

### 2.3. Assessment Instruments

The instruments used were a questionnaire for sociodemographic characterisation, the Positive Mental Health Questionnaire and the Mental Health Knowledge Questionnaire, all in an online format.

#### 2.3.1. PMHQ

The multifactorial model created by Teresa Lluch served as the foundation for the PMHQ used in this study [16]. The basis of this model is the evaluation of 6 factors that underpinned the construction of Teresa Lluch’s PMH assessment instrument translated and validated for the Portuguese population by Sequeira and Carvalho [18]. The questionnaire comprises 39 items assessed on an ordinal scale, with a maximum score of 156 and a minimum of 39 points, distributed unequally among the 6 factors: (1) personal satisfaction; (2) pro-social attitude; (3) self-control related to emotional balance; (4) autonomy referred to as the capacity of the person to make decisions; (5) problem solving and self-actualization and (6) interpersonal relationship skills associated with the ability to communicate with others and develop harmonious interpersonal relationships.

The questions come in the form of affirmative or negative statements, and the answers range from always or almost always, quite often, sometimes to never or rarely on a scale of 1 to 4. The questionnaire provides both individual scores for each element as well as an overall PMH score (total of item scores). The PMH level is therefore higher for higher values achieved using this assessment tool.

The original questionnaire was validated in a student population and the general population, yielding an internal consistency (Cronbach’s alpha) between 0.89 and 0.90 and a test–retest correlation of 0.85.

#### 2.3.2. MHKQ

The Mental Health Knowledge Questionnaire, translated and validated for the Portuguese population by Chaves, Sequeira, and Duarte [19], is divided into three parts. The first part has 16 statements that assess the knowledge of the characteristics of MH and mental disorders and beliefs in their epidemiology and are rated on a Likert-type scale in which 1 = “I totally disagree”; 2 = “I partially disagree”; 3 = “Neither agree nor disagree”; 4 = “I partially agree” and 5 = “I totally agree”. The total scores range from 14 to 70, and higher scores indicate higher levels of literacy. The second part consists of 4 items (from 17 to 20) that evaluate the activities to promote MH and are scored with 1 or 0 points for answers of “yes” and “no”, respectively. Finally, the third part has 10 items that assess what is important for good MH. Each item is rated on a 5-point Likert-type scale ranging from 1, “completely wrong”, to 5, “completely correct”, plus 0, “don’t know”. The global score ranges from 10 to 50 points and the interpretation of the results should be that the higher the score, the higher the levels of good MH. The instrument shows good psychometric adequacy attributes of both validity and reliability (Cronbach’s alpha coefficients ranging from 0.57 to 0.73 and a 2-week test–retest reliability of 0.68) and the measure may be considered adapted and validated for the Portuguese context. 

### 2.4. Statistical Analysis

A descriptive analysis of the sample was performed using frequency tables (in the case of qualitative variables) and the mean, maximum, minimum and standard deviation (in the case of quantitative variables). All dimensions under study were analysed for internal reliability using Cronbach’s alpha and all scales and subscales were transformed into percentages to facilitate comparisons and interpretations.

The Student’s *t*-test for two independent samples was used to detect the existence of statistically significant differences in the PMH, the MH literacy and the assessment of MH knowledge dimensions. The assumption of population normality was safeguarded under the central limit theorem. To study the interdependence between the different dimensions, a matrix with Pearson’s correlation coefficients and respective statistical significance was replicated; a *p* value < 0.05 was considered statistically significant. No tests of normality/homoscedasticity were conducted and normality was assumed. The statistical analysis of this study was performed using IBM SPSS Statistics 24.0 (Chicago, IL, USA).

## 3. Results

The social and labour characteristics of the 770 respondents in the sample are presented in Table 1. Most of the respondents were women (*n* = 495), with an average age of 36.3 years old. As many as 312 (40.5%) of the respondents had completed secondary education, showing an intermediate level of schooling. Also, most of the respondents were employed (64.6%; *n* = 498) (Table 1).

Similarly, most of the respondents (69.2%, *n* = 533) were (at the time of the questionnaire) or had been in quarantine or social isolation due to suspicion of COVID-19. Only 5.5% (*n* = 42) of the respondents indicated having an MH problem, but 7.3% (*n* = 56) stated that they took regular medication for an MH problem. It was found that 21.2% (*n* = 163) of the respondents had family members with mental illnesses.

When asked whether they believed that they slept enough hours for their needs, 61.3% (*n* = 472) answered affirmatively, and only 9.1% (*n* = 70) indicated taking sleeping medication.

More than half of the respondents (56.0%, *n* = 339) did not exercise. It was found that 81.4% (*n* = 628) of the respondents considered their diet to be healthy (Table 2).

The average PMH score (total scale) was 84.7% (standard deviation of 9.9%). The most positive contributions to this score came from the dimensions “self-control” and “personal fulfilment”. The mean score of the Good Mental Health Scale was 90.6% (standard deviation of 10.6%). The lowest mean scores were found in the assessment of knowledge on mental health with emphasis on the belief in the epidemiology of mental disorders and the awareness of health promotion activities.

Most of the scores show good internal reliability, except for the dimensions of the assessment of knowledge in MH which show weaker Cronbach’s alpha results (0.6 to 0.7) and the PMH pro-social attitude with a result very close to unacceptable (0.555) (Table 3).

The two variables “knowledge” and “literacy and good Mental Health” are both directly related to the variation in PMH, i.e., those respondents who had more knowledge/literacy concerning mental health (total score) and those who had better total scores for good mental health also presented higher values of PMH (total) (positive correlation with *p*-values < 0.001, despite the correlation values being 0.124 and 0.227, which represents a small effect) (Table 4).

From analysing Table 5, it can be seen that those who believe that they sleep enough hours for their needs show significantly higher PMH results.

Practising physical exercise was associated with higher scores for PMH and having a greater awareness of activities to promote MH. These findings are statistically significant (Table 6).

Eating food considered healthy meant having higher scores for PMH and having a greater awareness of activities to promote MH. These findings were statistically significant (Table 7). It should be mentioned that for the first time, the PMH dimension “Pro-social attitude” was a differentiating criterion between groups.

## 4. Discussion

The main objective of this study was to identify health behaviours impacting PMH during a pandemic phase and to verify if there was a relationship between literacy and PMH to prevent or try to minimize the impact of the pandemic on MH.

Based on our literature search, several studies reveal that the pandemic has had or will have a negative impact on the mental health of the general population [20,21,22,23]. A recent study conducted on patients, informal caregivers and health professionals revealed that depression, anxiety, post-traumatic stress symptoms, sleep disturbances, low self-esteem and decreased self-control are prevalent symptoms in individuals subjected to preventive measures such as quarantine and social isolation [20].

In other investigations conducted during the COVID-19 pandemic, some similar findings were observed regarding the specific aspects of PMH, specifically in the dimensions of self-control and personal achievement. Numerous studies on the self-control dimension revealed that it acts as a protective buffer between the perceived severity of the COVID-19 infection and MH, with those with the least amount of self-control being the most vulnerable [24,25].

Concerning the pro-social attitude dimension, the literature shows that all acts performed for the benefit of others offer a promising, flexible and low-cost intervention that can provide small but lasting benefits such as life satisfaction and well-being and consequently, promote better mental health. The emotional benefits suggest that pro-social attitudes can be used as interventions to improve MH [26,27,28].

The average level of happiness and life satisfaction during the pandemic was significantly lower than it was during the pre-pandemic period, according to studies on personal fulfilment [29,30]. Therefore, we can conclude that as these dimensions of PMH rise, the issues with MH become less severe. Thus, comparing the mean score of 84.7% for total scale for PMH in our study with that of another study carried out with Portuguese and Spanish nursing students using the same assessment instrument, we can affirm that our PMH values were also high during the pandemic phase [31] which corroborates our study.

Regarding the behaviours that promote good mental health, several authors (with different samples) concluded that getting enough sleep, eating well and practising physical exercise contributed significantly to maintaining or improving mental health during the pandemic phase [32,33,34].

Despite showing consensual results, some studies show that the number of individuals with sleep problems has increased since the beginning of the pandemic, with this variation being directly related to the increase in anxiety levels [35,36]. Thus, several authors defend practising good sleep hygiene as a coping strategy during a pandemic, namely in phases of social isolation [30].

Regarding the relationship between physical exercise and PMH levels, we found recent studies that are in line with our findings. These studies concluded that conscious promotion of physical activity and PMH is an effective strategy to reduce the negative impacts of the pandemic outbreak on both MH and physical health [36,37]. Of further note, another study conducted on women in China during the COVID-19 outbreak also revealed exercise as a common protective factor in depression, anxiety and stress [38]. However, it is important to note that the preventive measures against COVID-19, such as prolonged isolation and quarantine, make individuals prone to sedentary lifestyles that promote decreased physical activity. Thus, combating the sedentary lifestyle imposed by the pandemic by promoting physical activity and healthy eating can contribute to an eventual rebalancing of both physical and mental health [39].

It is also worth noting that a study conducted on Chinese university students revealed that both sleep and exercise are beneficial for mental health and probably regulate mental health through an interactive compensation mode [40].

In summary, we can infer that PMH is directly related to healthy lifestyles. Thus, in corroboration with the various studies found, we can say that promoting salutogenic behaviours also benefits MH.

Concerning health literacy, our assessment of knowledge on MH had the lowest mean scores with an emphasis on the belief in the epidemiology of mental disorders and the awareness of health promotion activities. Similar to our results, several studies in different populations (Saudi nursing students, Portuguese undergraduate students) during the pre-pandemic and pandemic phases obtained lower scores in the knowledge assessment domain. However, contrary to what we found, those studies also found higher values within literacy, namely in the belief in the epidemiology of mental disorder domain [41,42].

A recent study conducted in Italy showed that health literacy is an important factor that motivates people to follow preventive measures and consequently, adopt healthy lifestyles and behaviours [43]. Some authors reinforce that the PMH of the population worsened during the pandemic phase and that the focus of health services should be on strengthening resilience and health literacy [44]. Thus, studies infer that PMH literacy positively influences the status of MH in general [41] and is even considered a key determinant of PMH and therefore crucial for MH.

The results found in this study, in line with the other research cited, reinforces the conceptual aspects of positive mental health that have already been explored by other studies and expanding new lines of study. On the other hand, at a practical level, the correlations obtained open up new perspectives for the approach of care in the promotion of positive mental health. Due to differences in group sizes, there is a likelihood of type I errors. The probability of false positives is reduced because almost all relevant data were found to have *p* value less than 0.01.

### 4.1. Limitations and Suggestions for Future Studies

The results could be generalized if this study were to be repeated with specific work groups and in various geographic locations throughout Portugal and Europe. This study was carried out in a region of Portugal where the population is largely elderly and lacks access to computers and the internet. Given that the population tested was significantly larger than our sample size, we view this as a restriction.

We believe that the unequal composition of the sample in terms of gender could be a limitation in this study since more than half of the respondents were women (64.3%, *n* = 495) and that this is an important variable, particularly given the greater vulnerability of women in crisis/pandemic situations.

We also believe that further studies are needed to empower health professionals and the general population, with the main objective being to increase positive mental health, particularly in pandemic situations.

### 4.2. Practical Recommendations

The psychological consequences of the pandemic have had a major impact on the mental health of those affected, which is why we consider it essential to add a section with practical recommendations. In crisis/pandemic situations, it is essential that the population adopts healthy and preventive behaviours in order to mitigate mental health problems and at the same time, promote positive mental health. In this way, this study corroborates recent studies which tell us that the adoption of healthy behaviours such as regular physical exercise (preferably in open spaces), good sleep hygiene and healthy eating habits are fundamental.

Managing mental health and psychosocial well-being during a time of crisis is crucial for maintaining not only mental health but also physical health.

## 5. Conclusions

With this study, we can conclude that during the pandemic phase, those who had more knowledge of mental health and adopted healthy/health-promoting behaviours were the ones who showed better positive mental health. With the data from this study, we can infer that those who, during the pandemic phase, had more knowledge about mental health and adopted healthy/health-promoting behaviours were the ones who apparently had better positive mental health (Pearson’s coefficient = 0.124 and 95% confidence interval).

Continuous evaluation of the impact of changes in lifestyle behaviours associated with the pandemic is necessary and health promotion strategies aimed at the adoption/maintenance of positive health-related behaviours should be used to address the increase in mental illnesses during identical phases. Understanding the prevalence of behaviours promoting good mental health can provide key information on essential areas of intervention concerning health promotion/literacy programs. Thus, the results of this study may have implications for planning strategies to promote healthy behaviours, providing a theoretical basis for governments, health institutions and health professionals to implement or improve effective policies and interventions for better adaptation in future pandemic crises.

## Figures and Tables

**Table 1 behavsci-13-00845-t001:** Social and labour characterization of the sample.

	*N*	%
Sex		
Female	495	64.3
Male	275	35.7
Marital Status		
Single	339	44.0
Married	376	48.8
Divorced	51	6.6
Widow(er)	4	0.5
Level of Education		
Primary Education 1st Cycle	13	1.7
Primary Education 2nd/3rd cycle	37	4.8
Secondary Education	312	40.5
Bachelor’s Degree	20	2.6
Licentiate	300	39
Master’s Degree	80	10.4
Doctorate Degree	8	1
Employment Situation		
Employed	498	64.6
Employer/Entrepreneur	70	9.1
Unemployed	41	5.3
Retired	7	0.9
Student	154	20
Hourly Rate		
Less than 35 h per Week	146	19
35 h per Week	255	33.1
40 h per Week	218	28.3
More than 40 h per Week	151	19.6

**Age**: mean 36.31 years (SD 12.32); min: 18; max: 74.

**Table 2 behavsci-13-00845-t002:** Health behaviour.

	*N*	%
Are you or have you been in quarantine or social isolation due to suspicion of COVID-19?		
Yes	533	69.2
No	237	30.8
Do you have a mental health problem?		
Yes	42	5.5
No	728	94.5
Do you have or have you had family members with mental illnesses?		
Yes	163	21.2
No	607	78.8
Do you think you get enough sleep for your needs?		
Yes	472	61.3
No	298	38.7
Do you take sleeping medication?		
Yes	70	9.1
No	700	90.9
Do you regularly take medication for any mental health problems?		
Yes	56	7.3
No	714	92.7
Do you exercise?		
Yes	339	44.0
No	431	56.0
Do you consider your diet to be healthy?		
Yes	628	81.6
No	142	18.4

**Number of hours of sleep per day:** mean: 6.91 (1.24); min: 3; max: 12.

**Table 3 behavsci-13-00845-t003:** Descriptive analysis of the Positive Mental Health Scales and Sub-Scales, Mental Health Knowledge and Good Mental Health.

	Range (Min–Max)	Average	Standard Deviation	Average	Standard Deviation	Cronbach’s Alpha
Total Positive Mental Health (TPMH)	63–156	132.1	15.4	84.7	9.9	0.935
Personal Satisfaction (PMH1)	9–32	26.5	4.5	82.9	14.0	0.840
Pro-Social Attitude (PMH2)	11–20	18.2	1.8	91.0	9.1	0.555
Self-Monitoring (PMH3)	5–20	15.8	2.9	79.1	14.7	0.814
Autonomy (PMH4)	5–20	16.1	3.1	80.3	15.5	0.800
Problem Solving and Personal Achievement (PMH5)	15–36	31.6	3.7	87.8	10.2	0.788
Interpersonal Relationship Skills (PMH6)	11–28	23.8	3.3	85.1	11.8	0.743
Evaluation of Mental Health Knowledge						
Knowledge of the Characteristics of MH and Mental Disorders (ACS1)	13–25	21.8	2.4	87.3	9.5	0.677
Belief in the Epidemiology of Mental Disorders (ACS2)	6–25	13.4	3.7	53.5	14.6	0.699
Awareness of Health Promotion Activities (ACS3)	0–4	2.6	1.3	64.0	32.7	0.666
Good Mental Health Scale (GMH)	10–50	45.8	4.8	90.6	10.6	0.827

**Table 4 behavsci-13-00845-t004:** Correlation between Positive Mental Health, Mental Health Knowledge and Good Mental Health Scales (*n* = 770).

		SPMH	SMHK
SMHK	Pearson Correlation	0.124 **	
	*p*	0.001	
SGMH	Pearson Correlation	0.227 **	0.077 *
	*p*	0.000	0.033

** Correlation is significant at the 0.01 level (2-tailed). *. Correlation is significant at the 0.05 level (2-tailed). SPMH—sum of Positive Mental Health. SMHK—sum of Mental Health Knowledge. SGMH—sum of Good Mental Health.

**Table 5 behavsci-13-00845-t005:** Positive Mental Health, Mental Health Knowledge and Good Mental Health Scales according to sleep hygiene.

Do You Think You Get Enough Sleep for Your Needs?	Yes		No		
	Average	Standard Deviation	Average	Standard Deviation	*p*
Total Positive Mental Health (TPMH)	85.7	9.8	82.9	9.9	<0.001
Personal Satisfaction (PMH1)	84.6	13.8	80.1	14.0	<0.001
Pro-Social Attitude (PMH2)	91.0	9.1	91.1	9.3	0.946
Self-Monitoring (PMH3)	80.7	14.7	76.8	14.4	<0.001
Autonomy (PMH4)	82.3	14.8	77.2	16.1	<0.001
Problem Solving and Personal Achievement (PMH5)	88.6	9.7	86.5	10.7	<0.05
Interpersonal Relationship Skills (PMH6)	85.7	11.5	84.1	12.1	0.082
Knowledge of the Characteristics of MS and Mental Disorders (ACS1)	87.0	9.5	87.7	9.6	0.280
Belief in the Epidemiology of Mental Disorders (ACS2)	53.7	15.1	53.1	13.8	0.580
Awareness of Health Promotion Activities (ACS3)	64.3	33.2	63.6	32.1	0.770
Good Mental Health Scale (GMH)	90.4	10.5	91.0	10.7	0.473

**Table 6 behavsci-13-00845-t006:** Positive Mental Health, Mental Health Knowledge and Good Mental Health Scales according to the practice of physical exercise.

Do You Exercise?	Yes		No		
	Average	Standard Deviation	Average	Standard Deviation	*p*
Total Positive Mental Health (TPMH)	86.5	8.9	83.2	10.5	<0.001
Personal Satisfaction (PMH1)	85.7	12.6	80.7	14.7	<0.001
Pro-Social Attitude (PMH2)	91.0	8.9	91.0	9.3	0.996
Self-Monitoring (PMH3)	81.9	13.5	76.9	15.2	<0.001
Autonomy (PMH4)	83.1	13.4	78.1	16.7	<0.001
Problem Solving and Personal Achievement (PMH5)	89.4	9.6	86.6	10.4	<0.05
Interpersonal Relationship Skills (PMH6)	86.2	10.9	84.2	12.4	0.082
Knowledge of the Characteristics of MH and Mental Disorders (ACS1)	86.9	9.6	87.5	9.5	0.335
Belief in the Epidemiology of Mental Disorders (ACS2)	54.3	15.2	52.8	14.1	0.148
Awareness of Health Promotion Activities (ACS3)	69.2	31.4	59.9	33.2	<0.001
Good Mental Health Scale (GMH)	90.5	10.0	90.7	11.0	0.853

**Table 7 behavsci-13-00845-t007:** Positive Mental Health, Mental Health Knowledge and Good Mental Health Scales according to healthy eating.

Do You Consider Your Diet to Be Healthy?	Yes		No		
	Average	Standard Deviation	Average	Standard Deviation	*p*
Total Positive Mental Health (TPMH)	85.8	9.4	79.5	10.7	<0.001
Personal Satisfaction (PMH1)	84.7	13.1	75.0	15.4	<0.001
Pro-Social Attitude (PMH2)	91.6	8.7	88.5	10.6	<0.05
Self-Monitoring (PMH3)	80.6	14.0	72.6	15.8	<0.001
Autonomy (PMH4)	81.5	14.5	75.1	18.5	<0.001
Problem Solving and Personal Achievement (PMH5)	88.7	9.7	83.7	11.0	<0.001
Interpersonal Relationship Skills (PMH6)	86.0	11.3	80.9	13.0	0.082
Knowledge of the Characteristics of MH and Mental Disorders (ACS1)	87.3	9.6	86.9	9.3	0.655
Belief in the Epidemiology of Mental Disorders (ACS2)	53.7	14.7	52.4	14.1	0.339
Awareness of Health Promotion Activities (ACS3)	65.7	32.6	56.5	32.6	<0.05
Good Mental Health Scale (GMH)	90.8	10.6	89.7	10.6	0.235

## Data Availability

Data is available on request to the corresponding author.

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
