# Peer review of "COVID-19—Evidence of the Impact of Literacy and Salutogenic Behaviours in Positive Mental Health: A Cross-Sectional Study"

_behavsci, 2023, doi:10.3390/bs13100845_

Round 1

Reviewer 1 Report

Well-done for completing a complex survey and analysis of data. 

This is a cross-sectional survey among 770 Portuguese young mostly female population with limited general literacy and were employed (we do not know where) who had a positive covid diagnosis during the peak of the pandemic in Europe, and 69% were in quarantine. The findings are based on self-report with respect to the respondents' sleep pattern, diet, and exercise, and capturing their positive mental health traits and knowledge using two previously validated instruments, for one of which the authors have cited data, but not the other one (MHKQ did not have validation data). 

Several points:

- First of all, it would be necessary for the authors to show the factor structure of the instruments in the sample that they have not done. If they have that information, it needs to be reported, or perhaps as a separate paper or at least in this paper. Results of a measurement model of BOTH the instruments need to be presented.

- Second, this being a cross-sectional survey and with weak correlation, the conclusion needs to be tempered and softened. Perhaps suggest that health promoting behaviours (sleep, diet, exercise) are weakly correlated with positive mental health (Pearson's Coefficient = 0.124, what is the 95% Confidence Interval??) and there needs to be a discussion of possible confounders and the 95% confidence interval needs to be cited

- Third, given the nature of the questionnaire with ordinal responses, it is arguable if a summated Pearson Correlation Coefficient is appropriate or whether something like polychromic correlations would be more appropriate, so a comparison between the two approaches might be useful for the authors to consider. 

- The dispersion diagrams are better represented as box plots rather than scatterplots because there are staggered points and the variables are not strictly continuous. 

- It is recommended that the discussion section begins with a summary of the main findings and then proceed with the similarities and differences from prior research on the topic and some explanations and limitations of this study, noting that for a cross-sectional survey there are issues around reverse causality.

No major issue with the language.

Author Response

Thank you very much for your suggestions, questions and requests for clarification. The present study will be significantly improved with your contributions, for which we are very grateful. We also appreciate the precious time you spent on your review.

Well-done for completing a complex survey and analysis of data. 

This is a cross-sectional survey among 770 Portuguese young mostly female population with limited general literacy and were employed (we do not know where) who had a positive covid diagnosis during the peak of the pandemic in Europe, and 69% were in quarantine. The findings are based on self-report with respect to the respondents' sleep pattern, diet, and exercise, and capturing their positive mental health traits and knowledge using two previously validated instruments, for one of which the authors have cited data, but not the other one (MHKQ did not have validation data).

- First of all, it would be necessary for the authors to show the factor structure of the instruments in the sample that they have not done. If they have that information, it needs to be reported, or perhaps as a separate paper or at least in this paper. Results of a measurement model of BOTH the instruments need to be presented.

The following text has been added to the data of the MHKQ instrument:

“Cronbach's alpha coefficients ranging from 0.57 to 0.73 and a 2-week test-retest reliability of 0.68.”

MHKQ was translated and validated for the Portuguese population and can be found in Chaves, C.; Sequeira, C.; Duarte, J. Questionário de Conhecimento de Saúde Mental, Escola Superior de Enfermagem do Porto, 2019.

Positive Mental Helaht questionnaire was translated and validated for the Portuguese population and can be found in Sequeira, Carlos; Carvalho, José Carlos; Sampaio, Francisco; Sá, Luís; Lluch-Canut, Teresa; Roldán-Merino, Juan - Avaliação das propriedades psicométricas do Questionário de Saúde Mental Positiva em estudantes portugueses do ensino superior. Revista Portuguesa de Saúde Mental. ISSN 1647-2160. (2014) Edição: 11, p. 45-53

- Second, this being a cross-sectional survey and with weak correlation, the conclusion needs to be tempered and softened. Perhaps suggest that health promoting behaviours (sleep, diet, exercise) are weakly correlated with positive mental health (Pearson's Coefficient = 0.124, what is the 95% Confidence Interval??) and there needs to be a discussion of possible confounders and the 95% confidence interval needs to be cited

Thank you very much for your inputs. The first paragraph of the conclusions has been changed to:

With the data resulting from this study, we can infer that those who, during the pandemic phase, had more knowledge about mental health and adopted healthy/health-promoting behaviours were the ones who apparently had better Positive Mental Health  (Pearson's Coefficient = 0.124 and  95% Confidence Interval).

- Third, given the nature of the questionnaire with ordinal responses, it is arguable if a summated Pearson Correlation Coefficient is appropriate or whether something like polychromic correlations would be more appropriate, so a comparison between the two approaches might be useful for the authors to consider.

Thanks for your suggestion. This made us reflect on the direction of our research and the statistical analyzes we have carried out. We considered in depth these questions. Furthermore, we presented this question to the statistician who worked together with our team. Its justification is that the population follows a normal distribution and, despite the instruments using an ordinal scale and, therefore, not continuous, their corresponding classifications are, and we can use Pearson Correlation. However, this note will be recorded and in future studies we will take this issue into consideration.

- The dispersion diagrams are better represented as box plots rather than scatterplots because there are staggered points and the variables are not strictly continuous.

Thanks for your suggestion. However, in agreement with another reviewer, we have chosen to delete the graphics from this article.

- It is recommended that the discussion section begins with a summary of the main findings and then proceed with the similarities and differences from prior research on the topic and some explanations and limitations of this study, noting that for a cross-sectional survey there are issues around reverse causality.

Regarding the limitations of the study, we understand and accept your suggestion. We have made the change in section 4.1. Limitations and suggestions for future studies. We have therefore added the following text:

We believe that the unequal composition of the sample in terms of gender could be a limitation in this study, since more than half of the respondents were women (64.3%, n=495) and considering that this is an important variable, particularly given the greater vulnerability of women in crisis/pandemic situations.

The following text was added at the beginning of the discussion:

After analysing the data, we can infer that there appears to be a relationship between positive mental health, health literacy and the adoption of health-promoting behaviours. Thus, those who have more health knowledge and healthy lifestyle behaviours are able to mitigate mental health problems and improve their positive mental health.

Reviewer 2 Report

This paper addresses the relationship between some aspects of mental health and variables such as health literacy and salutogenic behaviors.

The theoretical framework is well laid out, with updated bibliographic references as is the case in the entire body of research generated from the pandemic.

With respect to the theoretical model chosen, the authors speak of two opposing models, that of Keyes (which they describe as predominant) and Lluch's factorial model (lines 50-65). A better justification of the choice of model for the present study would be enlightening.

In reference to the study itself, the correlational methodology limits certain causal inferences. The authors are cautious at some points in the work, in which they speak of "relationships", although the very objective of the study, as it is written in lines 225-227, is ambiguous in this sense. The methodological approach (descriptive and correlational analyses in a sample with a certain diversity of gender, age or possible infection of the disease with self-reported measures) calls for caution.

Thus, for example, fear of disease and contagion has already been duly pointed out as a mediating variable in the mental health measures of the population. On the other hand, taking into account the composition of the sample, unequal in terms of gender, and knowing the differential effect that this variable has had on the population during the emotional responses in the pandemic, it is missing some mention, even if only in the limitations section, of a gender approach in the research through, for example, specific analyses of the weight of this variable.

Finally, given that one of the focuses of the paper is on the study of salutogenic factors, I think the reader would appreciate a section, however brief, of practical recommendations derived from the findings of the study.

Author Response

Thank you very much for your suggestions, questions and requests for clarification. The present study will be significantly improved with your contributions, for which we are very grateful. We also appreciate the precious time you spent on your review. This way we hope to be able to answer all your suggestions accurately.

With respect to the theoretical model chosen, the authors speak of two opposing models, that of Keyes and Lluch's factorial model (lines 50-65). A better justification of the choice of model for the present study would be enlightening.

Thank you very much for your question and request for clarification.

We decided to choose Lluch's Multifactorial Model of PMH because it enables a salutogenic assessment of Mental Health and allows health professionals to be health promoting agents by implementing interventions capable of promoting satisfaction, a pro-social attitude, self-control, autonomy, problem-solving skills and interpersonal relationship skills.

The following text has been added:

This model allows for a salutogenic assessment of mental health and enables health professionals to be health-promoting agents by implementing interventions capable of promoting satisfaction, a pro-social attitude, self-control, autonomy, problem-solving skills and interpersonal relationships.

In reference to the study itself, the correlational methodology limits certain causal inferences. The authors are cautious at some points in the work, in which they speak of "relationships", although the very objective of the study, as it is written in lines 225-227, is ambiguous in this sense. The methodological approach (descriptive and correlational analyses in a sample with a certain diversity of gender, age or possible infection of the disease with self-reported measures) calls for caution.

Thus, for example, fear of disease and contagion has already been duly pointed out as a mediating variable in the mental health measures of the population. On the other hand, taking into account the composition of the sample, unequal in terms of gender, and knowing the differential effect that this variable has had on the population during the emotional responses in the pandemic, it is missing some mention, even if only in the limitations section, of a gender approach in the research through, for example, specific analyses of the weight of this variable.

Thank you very much for your suggestion. We fully agree with your analysis and have therefore added the following text to the limitations section:

We believe that the unequal composition of the sample in terms of gender could be a limitation in this study, since more than half of the respondents were women (64.3%, n=495) and considering that this is an important variable, particularly given the greater vulnerability of women in crisis/pandemic situations.

Finally, given that one of the focuses of the paper is on the study of salutogenic factors, I think the reader would appreciate a section, however brief, of practical recommendations derived from the findings of the study.

We appreciate and thank you for your suggestion. We agree that this information will be well received by readers. A sub-section has therefore been added:

4.2 - Practical Recommendations

The psychological consequences of the pandemic have had a major impact on the mental health of those affected, which is why we consider it essential to add a section with practical recommendations. In this way, and in crisis/pandemic situations, it is essential that the population adopts healthy and preventive behaviours in order to mitigate mental health problems and at the same time promote their positive mental health. In this way, this study corroborates recent studies which tell us that the adoption of healthy behaviours such as regular physical exercise (preferably in open spaces), good sleep hygiene and healthy eating habits are fundamental.

Managing mental health and psychosocial wellbeing during a time of crisis is crucial to maintaining not only your mental health but also your physical health.

Reviewer 3 Report

The article COVID-19 - Evidence of the Impact of Literacy and Salutogenic Behaviours on Positive Mental Health: A Cross-Sectional Study aims to investigate the influence of health literacy on the adoption of behaviours that foster Positive Mental Health during the COVID-19 pandemic.

This empirical article presents a compelling and positive perspective, though it requires some enhancements to unlock its full potential.

- The Abstract needs a specification of the method used

- Content: some ideas need more supoort on bibliography (e.g. paragraph 75-76

- Method:

Is not mentioned in the abstract, and that is very important omission.

Mention data & site recolection in desing, point out the pandemic situation at site of recolection  

The sample design is not very clear: maybe is more simple pointing: "opportunity sample".

Instruments: the PMHQ translation of Luch or the MHKQ of Chaves are only a traductions or validations?. it is not clear in the text. If the instrument does not have a Portuguese validation, please indicate this, and reflect about in disussion.

Some type of adaptation or validation was made for its online application?. Is there a version for online use?

Analysis:

- The purpose and rationale for using Cronbach analysis are unclear. What information does it contribute?

- Is not clear if your sample meets the assumptions of normality or homoscedasticity. Then, is unclear how T or Pearson can be used effectively.

- Hypotheses are unclear, and so is the grouping and the criteria. If you used multiple analyses to detect significant differences, you may need to apply a Bonferroni corre

- Significant differences in group sample sizes: Get enough sleep (61.3%), exercise regularly (56%), and maintain good health (81.6%). These significant differences may potentially lead to Type I errors, although there is no mention in the discussions.

Results

Authors place great importance on central tendency results. These results in the light of the methodology used are merely descriptive and not very robust, a simple summary table is better than a paragraph.

The usefulness of mentioning the raw or direct data and then also percentages and in both cases SD is not clear.

There is no reflection about correlation size, all the size are little and then whats are the importance of it?

Discussion:

It is too superficial since it does not discuss the study method, limitations and weaknesses. alternative explanations and reflections. It also does not address practical implications or future design improvements.

E.g. Are there any implications if the sample originates from a location where people are suspected of infections? Who peolple don't go there? And then, Who is not considered in your sample? 

E.g. Are there any risks of circular reasoning? Some factor of PMHQ and MHKQ are very close?... or sleep ehough/do exersices... is a close dimension in all the questionaries too.

Formal Issues

Are many errors in the cite format (eg. line 67, 109..

In tables, authors should report "n" and percentage separately in each column. For cognitive efficiency, it may be preferable to use "n(%)" in the same column. eg. 495(64,3%) 

In Table 3, be more synthetic. Use ONE column called range (min-max), use OR: the raw result or the percentages, but not both.

A dispersions diagrams is not required as it does not provide any information and is not aborded in the text

Author Response

Thank you very much for your suggestions, questions and requests for clarification. The present study will be significantly improved with your contributions, for which we are very grateful. We also appreciate the precious time you spent on your review.

The article COVID-19 - Evidence of the Impact of Literacy and Salutogenic Behaviours on Positive Mental Health: A Cross-Sectional Study aims to investigate the influence of health literacy on the adoption of behaviours that foster Positive Mental Health during the COVID-19 pandemic.

This empirical article presents a compelling and positive perspective, though it requires some enhancements to unlock its full potential.

- The Abstract needs a specification of the method used

The following text has been added:

A descriptive, cross-sectional and correlational study was conducted.

- Content: some ideas need more supoort on bibliography (e.g. paragraph 75-76)

Relevant bibliography was added to lines 75-76:

Kobau, R., Seligman, M. E. P., Peterson, C., Diener, E., Zack, M. M., Chapman, D., & Thompson, W. (2011). Mental Health Promotion in Public Health: Perspectives and Strategies From Positive Psychology. American Journal of Public Health, 101(8), e1–e9. https://doi.org/10.2105/AJPH.2010.300083

- Method:

Is not mentioned in the abstract, and that is very important omission.

Mention data & site recolection in desing, point out the pandemic situation at site of recolection  

The sample design is not very clear: maybe is more simple pointing: "opportunity sample".

The method, data & site recolection in desing and the pandemic situation at site of recolection has been added. The sample design was change for opportunity sample.

The following text was added in 2.1. Design and Participants:

“The opportunity sample…”

“…during the pandemic…”

Instruments: the PMHQ translation of Luch or the MHKQ of Chaves are only a traductions or validations?. it is not clear in the text. If the instrument does not have a Portuguese validation, please indicate this, and reflect about in disussion.

Both instruments had already been translated and validated. This information was corrected in the article:

“translated and validated for the Portuguese population”.

Some type of adaptation or validation was made for its online application?. Is there a version for online use?

Both instruments used were created and/or translated and validated by members of this research team. There was no need to specifically validate the instruments for their online use.

Analysis:

- The purpose and rationale for using Cronbach analysis are unclear. What information does it contribute?

The intention was to understand if the items were consistently measuring the same characteristics. For this reason, we chose to add Cronbach's alpha analysis to our study.

- Is not clear if your sample meets the assumptions of normality or homoscedasticity. Then, is unclear how T or Pearson can be used effectively.

We assumed that normality was safeguarded under the central limit theorem. With this in mind we applied the t-test.

- Hypotheses are unclear, and so is the grouping and the criteria. If you used multiple analyses to detect significant differences, you may need to apply a Bonferroni corre

Thank you very much for your suggestion. This question made us raise other research hypotheses that we will explore in the future. Regarding the use of the Bonferroni correction test, we chose to include it in future analyzes of the data of this study.

- Significant differences in group sample sizes: Get enough sleep (61.3%), exercise regularly (56%), and maintain good health (81.6%). These significant differences may potentially lead to Type I errors, although there is no mention in the discussions.

Thank you for your considerations. A way to minimize this probability is to understand that a false positive can occur more likely if p<0.05. This can be minimized if the significance level is 1%. Almost all relevant data was found to have p value less than 0.01. This information has been added to the discussion:

Due to differences in group sizes, there is a likelihood of type I errors. The probability of false positives is reduced because almost all relevant data was found to have p value less than 0.01.

Results

Authors place great importance on central tendency results. These results in the light of the methodology used are merely descriptive and not very robust, a simple summary table is better than a paragraph.

Thank you very much for your suggestion. We tried our best to summarize and compile the most important information. We prefer to keep the results as we presented them. However, if this is an imperative condition on the part of the reviewer, we will make an effort to change the way the data is presented to meet your preferences.

The usefulness of mentioning the raw or direct data and then also percentages and in both cases SD is not clear.

The idea of presenting the data in this way is to make it easier to read the data and give the readers more information. As we mentioned previously, if this is imperative condition on the part of the reviewer, we will make an effort to change the way the data is presented.

There is no reflection about correlation size, all the size are little and then whats are the importance of it?

The following text was added:

“…, despite the correlation values being 0.124 and 0.227, which represents a small effect.”

Discussion:

It is too superficial since it does not discuss the study method, limitations and weaknesses. alternative explanations and reflections. It also does not address practical implications or future design improvements.

E.g. Are there any implications if the sample originates from a location where people are suspected of infections? Who peolple don't go there? And then, Who is not considered in your sample?

E.g. Are there any risks of circular reasoning? Some factor of PMHQ and MHKQ are very close?... or sleep ehough/do exersices... is a close dimension in all the questionaries too.

Limitations were improved and a chapter about practical implications was added:

4.1. Limitations and suggestions for future studies

The replication of this study with specific work groups and in different geo-graphical areas in Portugal and in the European community would allow for the generalisation of results. This study was conducted in an area of Portugal where the population is very aged and many do not have computer resources (no computer or inter-net). We consider this a limitation because the population tested was much larger than our sample size.

We believe that the unequal composition of the sample in terms of gender could be a limitation in this study, since more than half of the respondents were women (64.3%, n=495) and considering that this is an important variable, particularly given the greater vulnerability of women in crisis/pandemic situations.

We also consider that further studies are needed to empower health professionals and the general population, with the main objective of increasing positive mental health, particularly in pandemic situations.

4.2. Practical recommendations

The psychological consequences of the pandemic have had a major impact on the mental health of those affected, which is why we consider it essential to add a section with practical recommendations. In this way, and in crisis/pandemic situations, it is essential that the population adopts healthy and preventive behaviours in order to mitigate mental health problems and at the same time promote their positive mental health. In this way, this study corroborates recent studies which tell us that the adoption of healthy behaviours such as regular physical exercise (preferably in open spaces), good sleep hygiene and healthy eating habits are fundamental.

Managing mental health and psychosocial wellbeing during a time of crisis is crucial to maintaining not only your mental health but also your physical health.

Formal Issues

Are many errors in the cite format (eg. line 67, 109..

Thank you for your observation. Citations were corrected.

In tables, authors should report "n" and percentage separately in each column. For cognitive efficiency, it may be preferable to use "n(%)" in the same column. eg. 495(64,3%)

The idea of presenting the data in this way is to make it easier to read the data and give the readers more information. As we mentioned previously, if this is imperative condition on the part of the reviewer, we will make an effort to change the way the data is presented.

In Table 3, be more synthetic. Use ONE column called range (min-max), use OR: the raw result or the percentages, but not both.

Thank you for your considerations. Range (min-max) was updated for only one column. Raw result was used.

A dispersions diagrams is not required as it does not provide any information and is not aborded in the text

Diagrams were deleted.

Round 2

Reviewer 1 Report

Thanks for the corrections. 

Author Response

We appreciate the time you took to review our article. We are very pleased that we have responded in accordance with what you suggested and agree that all your suggestions and requests for clarification have been very important and have significantly improved our article. Thank you.

Reviewer 3 Report

A positive step forward,

But, please ensure you add the number of participants in the abstract.

Additionally, clarify in section 2.1 Design and Participants whether the data was collected "during the pandemic's confinement phase or before vaccination" (e.g). Please be specific. 

In the Assessment section, please include the phrase "all in an online format for application" in the instrument description. Thank you.

Finally, please clarify in the Analysis section that no tests of normality/homoscedasticity were conducted and normality was assumed.

Incorporate the Bonferroni correction and specific test hypotheses into future research.

Additionally, improve the "Practical Recommendation" section by including advice on regular exercise, good sleep, hygiene and healthy habits. What contributions and practical implications does this study offer?

Formal aspect: E.g. a dot in line 110, see reference 19 y 20 (line 412-415)

Author Response

Once again, we'd like to thank you for the opportunity to improve our article with all your contributions.

Please ensure you add the number of participants in the abstract.

The following text has been added at the abstract:

"... in a sample of 770 patients and..."

Additionally, clarify in section 2.1 Design and Participants whether the data was collected "during the pandemic's confinement phase or before vaccination" (e.g). Please be specific. 

We appreciate your request for clarification and agree with the relevance of the question.

The following text has been added in section 2.1 Design and Participants:

“The data was collected during the lockdown phase of the pandemic and before vaccination.”

In the Assessment section, please include the phrase "all in an online format for application" in the instrument description. Thank you.

The text "all in an online format for application" was including in the Assessment section.

Finally, please clarify in the Analysis section that no tests of normality/homoscedasticity were conducted and normality was assumed.

In the Analysis section the following text has been added:

"No tests of normality/homoscedasticity were conducted and normality was assumed."

Incorporate the Bonferroni correction and specific test hypotheses into future research.

We appreciate and thank you for your suggestion, as we are considering using these methods in a future article that is already under construction. Thank you

Additionally, improve the "Practical Recommendation" section by including advice on regular exercise, good sleep, hygiene and healthy habits. What contributions and practical implications does this study offer?

Thank you very much for your suggestion. In the Conclusion section we write that “the results of this study may have implications for planning strategies to promote healthy behaviors, providing a theoretical basis for governments, health institutions and health professionals to implement or improve effective policies and interventions for better adaptation in future pandemic crises.”

Formal aspect: E.g. a dot in line 110, see reference 19 y 20 (line 412-415)

Formal aspects were corrected.

Thank you very much once again for the effort you put into reviewing this article. We are aware that your contribution was fundamental to improving its quality.